# Physicochemical Characterizations, Digestibility, and Lipolysis Inhibitory Effects of Highland Barley Resistant Starches Prepared by Physical and Enzymatic Methods

**DOI:** 10.3390/molecules28031065

**Published:** 2023-01-20

**Authors:** Cong Wang, Xinyi Tian, Xiayin Zhang, Zhiming Zhang, Xiaoyu Zhang, Xiaoxiong Zeng

**Affiliations:** 1College of Food Science and Technology, Nanjing Agricultural University, Nanjing 210095, China; 2College of Biosystems Engineering and Food Science, Zhejiang University, Hangzhou 310058, China

**Keywords:** highland barley, resistant starch, physical and enzymatic modifications, physicochemical characterization, lipolysis inhibitory effect

## Abstract

This study aimed to investigate the differences in the physicochemical and structural characteristics, digestibility, and lipolysis inhibitory potential in vitro of highland barley resistant starches (HBRSs) prepared by autoclaving (HBSA), microwave-assisted autoclaving (HBSM), isoamylase (HBSI) and pullulanase (HBSP) debranching modifications. Results revealed that the resistant starch content of native starch was significantly elevated after modifications. HBSA and HBSM showed distinctly higher swelling power and water-binding capacities along with lower amylose amounts and solubilities than those of HBSI and HBSP (*p* < 0.05). Fourier transform infrared spectroscopy and X-ray diffraction exhibited that HBSP displayed the highest degree of the ordered crystalline region and crystallinity with a mixture of C_B_- and V-type polymorphs. Meanwhile, HBSA and HBSM were characterized by their high degree of the amorphous region with a mixture of B- and V-type polymorphs. Physical and enzymatic modifications resulted in different functionalities of HBRSs, among which HBSP showed the lowest digestibility and HBSM exhibited the highest inhibitory activity on lipolysis due to their structure and structure-based morphology and particle size. This study provided significant insights into the development of native starch from highland barley as an alternative functional food.

## 1. Introduction

Highland barley (*Hordeum vulgare* L. var. *nudum* Hook. f., HB), commonly known as Qingke and Himalayan barley, is a traditional staple crop in Qinghai-Tibet Plateau region of China [1]. It provides abundant nutrients, such as soluble dietary fiber, beta-glucan, arabinoxylan, and polyphenols, which present potential health-promoting benefits, making it suitable to be developed into functional foods [2]. Starch is the major component of HB (49.14–68.62%) and composed of linear amylose and branched amylopectin (74.0–78.0%) with different proportions based on the cultivars and planting conditions [3]. The previous study demonstrated that the contents of rapidly digestible starch (RDS), slowly digestible starch (SDS), and resistant starch (RS) in HB starch (HBS) were around 96.19%, 1.54%, and 2.27%, whereas RDS was digested within 20.0 min and its long-term intake elevated the risk of metabolic disorder, causing hyperglycemia and hyperlipidemia [4,5,6]. RS is generally stable during digestion and enters the colon fermented, it is capable of improving lipid metabolism and has a prebiotic effect [7]. Therefore, promoting the formation of RS based on RDS can effectively reduce digestibility and endows the novel functionality to HBS.

Various methods have been employed in the last decade to improve RS amount, including heat–moisture, autoclaving–cooling, autoclaving–microwave, ultrahigh pressure treatment, enzyme debranching, and chemical derivations [8,9,10,11,12,13]. Among them, physical and enzymatic modifications are the most commonly used techniques due to their convenience and sustainability [14]. Autoclaving, a typical method to physically modify starch, can facilitate the degradation of the long-chain branch and gelatinization process, which enhances RS formation in subsequent retrogradation during the cooling or freezing process [15]. Microwave treatment usually induces the rapid depolymerization of starch granules through the breakage of glycosidic bonds and the rearrangement of crystalline regions [16]. Its combination with autoclaving–cooling treatment was suggested to be beneficial for promoting the formation of RS [11]. Debranching enzymes, like isoamylase and pullulanase, can specifically hydrolyze 1,6-α-D-glycosidic bonds at branching points, resulting in increased numbers of amylose and amylopectin with appropriate chain length, which contribute to the formation of RS through re-aggregating amylose into the double helix [12]. Nowadays, heat–moisture, roasting, superheated steam, and pullulanase-autoclaving have been successfully performed to promote the resistance of HBS from being digested [8,17,18]. Nevertheless, limited information is available about the differences in the preparation of highland barley resistant starches (HBRSs) by physical and enzymatic methods, which are illustrated to have distinct effects on the physicochemical characteristics and digestibility of starch [19,20].

Hyperlipidemia, characterized by elevated levels of cholesterol and triglycerides in the blood, plays a significant role in the development of obesity, diabetes, and cardiovascular disease [21]. Recent studies have shown that the consumption of RS was beneficial for the delay of fat absorption and utilization, satiety promotion, bile acid binding, and short-chain fatty acid generation, which are vital for maintaining lipid homeostasis [22,23,24]. To our knowledge, few studies have focused on the lipolysis inhibitory potential of HBRSs prepared by different methods.

In this study, four methods were used to prepare HBRSs, including autoclaving, microwave-assisted autoclaving, isoamylase, and pullulanase debranching. The physicochemical properties, digestibility, and inhibitory effects of lipolysis in vitro of HBRSs were further evaluated, which could facilitate the development of HBRSs as alternative functional foods and provide a basis for other starch modifications.

## 2. Results

### 2.1. Chemical Compositions of HBRSs Prepared by Different Methods

The contents of amylose, protein, and RS of HBRSs were summarized in Table 1. It was revealed that the amylose contents of HBSA and HBSM were reduced compared to that of HBS, with decreased rates of 25.62% and 11.31% (Appendix A), respectively. Contrarily, the enzyme debranching resulted in an evident increase in the amylose content of HBS, which was significantly higher than those of HBSA and HBSM (*p* < 0.05). Among all the HBRSs, HBSI showed the highest elevation in amylose amount, whereas those of HBSA and HBSM showed non-significant differences (*p* > 0.05). The RS contents in HBRSs ranged from 21.37% to 22.57%, with the highest value for HBSI and no apparent difference between physical and enzymatic modifications. Meanwhile, no protein was detected in HBRSs, suggesting that their digestibility was not affected by protein during the modifications [25].

### 2.2. Solubilities, Swelling Power, and Water-Binding Capacities of HBRSs Prepared by Different Methods

As depicted in Table 1, the solubilities of HBRSs ranged from 3.46% to 63.24%, in which HBSA and HBSM showed significantly lower values than HBS, whereas HBSI and HBSP exhibited noticeably higher values (*p* < 0.05), comparatively. HBSI revealed the highest solubility, and the minimum solubility was observed in HBSM. In contrast, the swelling power of HBSA and HBSM showed 26.67–77.03% higher values than those of HBSI and HBSP (*p* < 0.05), with the highest swelling power of HBSA. The autoclaving and microwave-assisted autoclaving modifications also led to prominent promotions in the swelling power of HBS (Appendix A). In addition, the modifications of HBS resulted in significant decreases in its water-binding capacity, with the reduction rates varying from 98.17% to 99.01%. It was noteworthy that the enzymatic modifications had higher adverse effects on the water-binding capacity of HBS than physical modifications (*p* < 0.05).

### 2.3. Morphological Properties of HBRSs Prepared by Different Methods

The native starch granules from HB (HBS) were observed to present a bimodal size distribution with the surface of the smooth and shaped lenticular (Appendix A), which was altered after modifications. At the magnification of 50.0×, the granular shape of HBS was observed to be entirely disrupted into polyhedral fragments with a rough and non-uniform surface (Figure 1). HBSI showed a distinct morphological difference compared with other modified HBS, revealing smaller particle sizes with a loose surface. At the magnification of 500.0×, a strip-like structure was found to cover the surface of HBSA, making it rougher than HBSM, featured by a compact and dense structure. In comparison, the surface of HBS after enzymatic modification appeared to be covered with randomly distributed cavities, particularly for HBSI. Further analysis at 3000.0× revealed that the appearance of HBS was similar within the same modification type and varied from that under other treatments.

### 2.4. FT-IR Spectra of HBRSs Prepared by Different Methods

The FT-IR spectra for HBRSs are shown in Figure 2A, revealing slight differences in their chemical groups compared to that of HBS (Appendix A). The characteristic broadband ranged from 3100.0 cm^−1^ to 3700.0 cm^−1^, and was generally attributed to the -OH stretching vibration of hydrogen bonds among intra- or inter-molecules [26]. Results showed that the absorption bands of HBSA and HBSM from 3100.0 cm^−1^ to 3700.0 cm^−1^ were narrower than that of HBS, indicating a more vital organization among the molecules that occurred during the gelatinization and retrogradation [27]. On the contrary, HBSI and HBSP showed relatively wider bandwidths than HBS, suggesting a higher crystallite structure [19]. It was noteworthy that the representative peak of -OH was found to be around 3370.0 cm^−1^, whereas the microwave-assisted autoclaving and isoamylase debranching modifications led to apparent redshifts with the absorption changed to 3392.0 cm^−1^, representing a substitution of the O-H bound [28]. No evident impacts on the -OH peak were found in HBSA and HBSP. From the spectra (Figure 2A), the most significant variations were observed from 1158.0 cm^−1^ to 575.0 cm^−1^. It was demonstrated that the characteristic absorptions at 995.0 cm^−1^ and 1047.0 cm^−1^ were associated with the order structure of starch, and that of 1022.0 cm^−1^ was related to its amorphous region [29]. Therefore, the intensity proportions at 995.0/1022.0 cm^−1^ and 1047.0/1022.0 cm^−1^ could be considered indicators for the degrees of the double helix and ordered crystalline region [19]. The proportions of 1047.0/1022.0 cm^−1^ and 995.0/1022.0 cm^−1^ of HBRSs were estimated to be 1.03 and 1.07 (HBSA), 1.06 and 1.11 (HBSM), 1.21 and 3.42 (HBSI), 1.94 and 2.15 (HBSP), separately. Compared with HBS (1.24 and 1.15), it could conclude that the enzymatic modifications were beneficial for promoting the degree of molecular order, whereas the physical modifications showed adverse effects. In addition, the absorption intensities of HBSI and HBSP around 2932.0 cm^−1^ and 1645.0 cm^−1^ were lower than those of HBSA and HBSM, indicating that the enzymatic branching resulted in more reductions of the C-H and C=O groups [30].

### 2.5. XRD Patterns of HBRSs Prepared by Different Methods

As shown in Appendix A, HBS exhibited firm intensity peaks at the 2*θ* diffraction angle around 15.0° and 23.0° coupled with a dual-peak around 17.0° and 18.0° (2*θ*), which were representative characteristics of A-type crystalline structure, consistent with the previous report [2]. After modifications (Figure 2B), all the HBRSs showed evident reflections around 20.0° (2*θ*), which reflected the formation of crystalline V-amylose-lipid complexes [11]. HBSM exhibited the highest intensity at 20.0° (2*θ*) than other HBRSs. Regarding the spectra of HBS modified by physical methods, the characteristic reflections around 5.60 (2*θ*) and 25.74 (2*θ*) were observed, suggesting the formation of a B-type crystalline structure, in accordance with the findings in the literature [17,31]. The enzyme debranching of HBS resulted in different XRD patterns. It was observed that HBSI was characterized by reflections around 7.60 (2*θ*) and 12.76 (2*θ*), which corresponded to a V-type crystalline pattern [32]. In contrast, HBSP showed a typical C_B_-type crystalline pattern as evidenced by its unique reflections around 15.12 (2*θ*), 17.96 (2*θ*), 22.19 (2*θ*), and 23.52 (2*θ*) [33]. Consequently, it could conclude that the physical modifications led to the transformation of the A-type crystalline structure of HBS to a mixture of B- and V-type polymorphs, whereas its debranching by isoamylase and pullulanase showed V-type and a mixture of C_B_- and V-type polymorphs. Furthermore, the relative crystallinities of HBRSs were estimated to be ranged from 7.0% to 27.78%, with the highest value for HBSP (Figure 2C). Autoclaving, microwave-assisted autoclaving, and isoamylase debranching modifications of HBS significantly decreased its relative crystallinity (*p* < 0.05, Appendix A), similar to the previous findings, which demonstrated that the retrogradation and enzymatic modification of native starch appeared to have adverse effects on its relative crystallinity [34]. Considering the non-significant differences in RS content among HBRSs, it was suggested that their relative crystallinities were not directly correlated with their RS amounts [35].

### 2.6. Digestibility of HBRSs Prepared by Different Methods In Vitro

Short-time salivary digestion appeared to have little influence on HBRSs digestion. As the digestion continued, their hydrolysis occurred, accompanied by the release of glucose with various patterns in different digestion phases (Figure 3). It was revealed that the glucose released from HBRSs during the gastric and intestinal digestion ranged from 2.94 mg to 7.30 mg and 4.46 mg to 9.86 mg (Figure 3A), respectively. Among all the HBRSs, HBSA and HBSM were more susceptible to the hydrolysis of digestion enzymes, whereas HBSI was merely sensitive to pancreatin. Noteworthily, HBSP released the least amount of glucose, indicating the highest stability during digestion. The estimated hydrolysis ratios of HBRSs after the entire digestion process were estimated to range from 10.66% to 24.29%, with the highest and lowest ratios of HBSP and HBSM (Figure 3B), significantly lower than that of HBS (*p* < 0.05, 68.72%). Moreover, the enzymatic debranching of HBS resulted in a higher resistance to digestion than when modified by physical methods.

### 2.7. Inhibitory Effects of HBRSs on the Microstructure of Lipid Emulsion

To fully understand the effects of HBRSs on lipid digestion, their intervention on lipolysis of lipid emulsion was further investigated in vitro. Results showed that the oil droplets stained by Nile red were homogeneously distributed in the emulsion with small particle sizes after salivary digestion, and the estimated area (EA) of the droplets in each group showed slight differences (Figure 4A). After gastric digestion, the stability of lipid emulsion was destroyed, and the droplets appeared to gather together. It was found that the EA of droplets under the influence of HBRSs was estimated to be varied from 67.85 ± 0.78 to 84.00 ± 1.27, which were 7.36–22.31% larger than that of the control (Figure 4B). Meanwhile, HBSA and HBSM showed significantly higher promotions on the aggregation of lipids than HBSI and HBSP (*p* < 0.05). Lipolysis mainly occurred in intestinal digestion, whereas HBRSs preserved the lipids from being hydrolyzed, as evidenced by their EA of oil droplets ranged from 28.55 ± 0.49 to 52.60 ± 0.57, apparently higher than that of the control (16.05 ± 0.35). Similarly, the physical modifications seemed to be more beneficial for improving the lipolysis inhibition of HBS than enzymatic modifications.

### 2.8. Inhibitory Effects of HBRSs on the Release of FFAs

The releases of FFAs were further assessed to provide a complementary evaluation of the lipolysis-inhibitory effects of HBRSs. Compared with the control group, HBRSs significantly reduced the releases of FFAs, with their inhibitory rates ranging from 46.16% to 73.91% (Figure 4C). Consistently, HBSA and HBSM showed distinctly higher inhibition on FFAs release than HBSI and HBSP (*p* < 0.05), in which HBSM revealed the most increased prevention on lipolysis whilst the inhibitory activities of HBSI and HBSP were similar to that of cellulose.

## 3. Discussion

To improve the stability and potential functionality of native starch from highland barley, autoclaving, microwave-assisted autoclaving, and enzymatic debranching were conducted to prepare HBRSs for the first time. The differences in their physicochemical properties, digestibility, and lipolysis inhibitory activities were also assessed. It was found that the amylose contents of HBSI and HBSP were significantly higher than those of HBSA and HBSM (*p* < 0.05), whereas their RS amounts showed slight differences (Table 1). Autoclaving was usually involved in decomposing the crystalline region and amorphous mass of native starch under elevated temperature and pressure, evidently promoting its gelatinization efficiency [36]. Upon completing the cooling process, the amylose molecules with suitable mass were re-associated and packed into a crystalline region stabilized by hydrogen bonding (retrogradation), resulting in the formation of RS [37]. During this process, large amylose amounts, long chain length of molecules, high autoclaving temperature, and low retrogradation temperature with long duration were considered essential for influencing RS formation [9]. Microwave irradiation could facilitate the gelatinization of starch in a short period and induce the generation of free radicals, which led to the hydrolyzation of α-1,6-glycosidic linkage of the amylopectin [38]. The extensive treatment of microwaves might also trigger the breakdown of the amylopectin chain, promoting the formation of amylose [20]. From the non-significant contents of amylose in HBSA and HBSM, it was deduced that the preliminary microwave treatment (640.0 W, 2.0 min) of HBS was not strong enough to break the main chain of its amylopectin. Noteworthily, the RS contents in HBSA and HBSM were also slightly different, in contrast to the previous report demonstrating that microwave assistance was beneficial for RS formation under autoclaving treatment in the absence of retrogradation [11]. This phenomenon suggested that the structural differences caused by physical modifications were insufficient to influence the retrogradation process in HBRSs formation. Although isoamylase and pullulanase participated in the hydrolysis of α-1,6-glycosidic bonds in amylopectin, their specificities for substrate were quite different [12]. It was demonstrated that isoamylase preferred to hydrolyze amylopectin with large molecular mass and required at least three α-1,4 linked glucose units, whereas pullulanase required at least two α-1,4 linked glucose units [39]. In the present study, isoamylase debranching resulted in a significantly higher amount of amylose than pullulanase debranching (*p* < 0.05), indicating that the unit of 6^3^-α-maltosyl maltotriose characterized the main structure of HBS. Additionally, the formation of RS under enzymatic debranching primarily depended on the proportion of short amylose and amylopectin with appropriate chain length, which determined the following association and realignment in establishing new crystalline structures upon retrogradation [37]. However, the excessive debranching of native starch might promote the release of short chains, which disturb the nucleation and propagation in crystallite formation, particularly for A-type starch [40]. As a result, the RS amounts in HBSI and HBSP showed minor differences despite the apparent differences in amylose contents.

The crystalline structure of starch was composed of ordered double helices arranged by amylopectin with short chains (crystalline region) and separated by amorphous lamellae [41]. Different modifications generally exerted distinct impacts on the disruption of the crystalline structure, and the subsequent retrogradation led to diverse patterns and degrees of newly formed crystals [37]. As revealed in Figure 2B, the A-type crystalline structure of HBS was altered to a mixture of B- and V-type polymorphs after physical modification, consistent with the previous findings in autoclaving, microwave, and their combined processing of starch, whereas the B-type crystalline structure was considered as the typical crystalline pattern for retrograded starch in most cases [17,42]. Meanwhile, HBSI and HBSP were characterized by a V-type and a mixture of C_B_- and V-type polymorphs, suggesting the potential association between amylose and lipid of starch [32]. The crystallinity was mainly attributed to the crystal size, number of crystalline regions, and double helices orientation within the crystalline domains [43]. Results showed that the relative crystallinity and proportion at 1047.0/1022.0 cm^−1^ of HBSP were significantly higher than those of HBSA, HBSM, and HBSI (*p* < 0.05, Figure 2A,C), indicating a higher-order degree in the crystalline region [29]. For HBSI, the relatively higher degree of double helices contributed more to its crystallinity. The structure-based morphology and functionality also confirmed the different patterns of decomposition and re-association between physical and enzymatic modifications. It was found that the swelling power and water-binding capacities of HBSA and HBSM were significantly higher than those of HBSI and HBSP (*p* < 0.05), whereas their solubilities showed a negative tendency (Table 1). According to the proportions at 995.0/1022.0 cm^−1^ and 1047.0/1022.0 cm^−1^, HBSA and HBSM showed relatively lower degrees of double helices and ordered crystalline regions, which might be beneficial for water absorption in their amorphous region, promoting their swelling and water-binding properties [20]. In contrast, the significantly higher solubilities of HBSI and HBSP were presumed to be associated with their ordered structure (Figure 1), enlarged cavities, and decreased particle sizes compared with the compact and dense structures of HBSA and HBSM [44].

Starch digestion usually occurs in the gastrointestinal tract, and its digestibility was widely used to assess the postprandial glucose response, especially essential for diabetes patients [4]. Typically, starch’s digestibility is estimated by calculating its hydrolysis index within 180.0 min under α-amylose digestion [45]. In the present study, a simulated gastrointestinal digestion model mimicking the natural digestion process of starch in vivo was established and used to evaluate the digestibility of HBRSs. It was reported that the granule size, amylose/amylopectin ratio, degrees of double helices and ordered crystalline regions, and the existence of amylose-lipid complexes were dedicated to the digestibility of starch [46]. Accordingly, the significantly lower hydrolysis rates of HBSI and HBSP compared with those of HBSA and HBSM might be ascribed to their relatively higher degrees of ordered crystalline regions (*p* < 0.05, Figure 3B), restricting the susceptibility of digestion enzymes. Small granule size induced sizeable relative surface area of HBSI elevated its exposure to digestion enzymes, which coupled with its lower crystalline degree led to a higher hydrolysis rate than that of HBSP [47]. No apparent differences in hydrolysis rate were observed in HBSA and HBSM, despite their remarkable improvement in reducing the digestibility of HBS.

Lipid digestion and absorption suppression were generally recognized as effective strategies for preventing hyperlipidemia and hypercholesterolemia [23]. The bulking effect, viscosity, binding capacity, and fermentation of RS were primarily responsible for its hypolipidemic benefit [48]. In the present study, HBRSs facilitated the aggregation of oil droplets and prevented them from being hydrolyzed by digestion enzymes (Figure 4A). It was observed that HBSA and HBSM revealed significantly higher inhibitory effects on lipolysis than HBSI and HBSP (*p* < 0.05), which might be ascribed to their relatively higher degree of the amorphous region, contributing to their swelling and binding with lipids [49]. In contrast, the ordered crystalline structures of HBSI and HBSP limited their interaction with lipids, whereas their sizeable relative surface area and rough appearance were beneficial for lipids-binding, which together were involved in the control of lipolysis. Considering the results of the digestibility of HBRSs, it could be concluded that the modifications of HBS by pullulanase debranching and microwave-assisted autoclaving were respectively favorable for reducing its digestibility and improving its inhibitory activity on lipolysis from the perspective of functionality. Additionally, the selection of modification method for HBS primarily depended on the desired targeted functionality.

## 4. Materials and Methods

### 4.1. Materials and Chemicals

Highland barley was purchased from Xinlvkang Food Co., Ltd. (Qinghai, China). α-amylose (3700.0 U/g), pullulanase (1000.0 U/mL), and lipase (20,000.0 U/g) were obtained from Solarbio Science and Technology Co., Ltd. (Beijing, China). Isoamylase and pepsin (1:30,000.0) were acquired from Ruixiang Biological Technology (Shanghai, China) and Yuanye Bio-Technology (Shanghai, China), separately. Pancreatin (1:4000.0) was obtained from Ryon Biological Technology (Shanghai, China). Test kits for determining glucose, amylose, and protein were obtained from Jiancheng Bioengineering Institute (Nanjing, China). All other chemicals and reagents were of analytical grade.

### 4.2. Preparation of Native Starch from Highland Barley

Native starch was prepared according to the previous report with minor modifications [50]. The ground highland barley was suspended in 0.45% sodium metabisulfite solution (*w*/*v*) at the ratio of 1:10 and stirred at 150.0 rpm overnight. After that, the suspension was sieved through a 100.0 mesh (150.0 μm) sifter to collect the filtrate. The remaining residues were extracted with sodium bisulfite solution for another 30.0 min and filtered to obtain the filtrate. The above procedure was repeated to achieve the filtrate for the third time. The filtrated fractions were combined and stirred for another 1.5 h after the addition of 0.1 M NaCl solution at the ratio of 9:1. Finally, the mixture was centrifuged at 3000.0× *g* for 10.0 min, and the white residues were gathered and air-dried at 50.0 °C after completely removing the top grey layer by distilled water washing and regarded as native starch (HBS) from HB.

### 4.3. Physical and Enzymatic Modifications of Native Starch from Highland Barley

#### 4.3.1. Preparation of Autoclaving Modified Resistant Starch (HBSA)

The autoclaving modified RS was prepared by the previous report with minor modifications [20]. Briefly, HBS was suspended in deionized water at the concentration of 60.0 mg/mL and treated at 121.0 °C (0.115 MPa) for 20.0 min. Then, the paste was cooled to ambient temperature and stored at 4.0 °C overnight to induce the retrogradation of starch. After centrifuging at 3500.0× *g* for 15.0 min, the residues were collected and air-dried at 50.0 °C and named HBSA.

#### 4.3.2. Preparation of Microwave-Assisted Autoclaving Modified Resistant Starch (HBSM)

The microwave-assisted autoclaving modified RS was prepared by suspending 10.0 g of HBS in 100.0 mL of deionized water and gelatinized at 640.0 W of microwave power for 2.0 min (P70D20L-HP3, Galanz, Guangdong, China) [51]. After cooling to ambient temperature, the paste was stored at 4.0 °C overnight. Similar centrifugation and air-drying were performed to obtain HBSM.

#### 4.3.3. Preparation of Enzyme-Debranched Starch

Starch debranched by isoamylase was prepared by the previous method with minor modifications [52]. Briefly, 5.0% of HBS suspension was gelatinized in a boiling water bath for 60.0 min with constant stirring. After equilibrating to 45.0 °C, the pH of the paste was adjusted to 3.5 with 0.5 M of HCl, and 0.5% of isoamylase was added to initiate the hydrolysis. The mixture was incubated for another 48.0 h with constant stirring, and anhydrous ethanol was added to terminate the enzymatic hydrolysis at the final concentration of 80.0%. Finally, the mixture was centrifuged at 4500.0× *g* for 15.0 min, and the residues were air-dried at 40.0 °C, sieved with a 100.0 mesh sifter, and named HBSI.

Starch debranched by pullulanase was prepared by suspending 10.0 g of HBS in 100.0 mL of acetic acid buffer (0.01 M, pH 5.5) and incubated in a boiling water bath for 15.0 min [12]. Then, the gelatinized starch was heated at 130.0 °C for 60.0 min and equilibrated to 55.0 °C before adding 4.0 U/g of pullulanase (*w*/*w* starch). The mixture was incubated for another 4.0 h, and the hydrolysate was precipitated by adding anhydrous ethanol with continuous stirring overnight. Subsequently, the mixture was centrifuged at 4500.0× *g* for 15.0 min, and the residues were air-dried at 40.0 °C, sieved with a 100.0 mesh sifter, and named HBSP.

### 4.4. Characterization of the Modified Starches

#### 4.4.1. Chemical Composition Analysis

The amylose and protein contents were measured by using commercial kits. The RS content was determined by the previous report [34].

#### 4.4.2. Morphological Property

The morphological properties of HBRSs were investigated by a scanning electron microscope (JSM-6390, JEOL, Tokyo, Japan). The starch powder was placed on the specimen holder with the help of double-sided adhesive tapes and coated with gold powder. Each sample was observed at an accelerating voltage of 3.0 kV with 50.0, 500.0, and 3000.0-fold magnifications.

#### 4.4.3. Fourier Transform Infrared (FT-IR) Spectroscopy

Infrared spectra of HBRSs were acquired by a Fourier transform infrared (FT-IR) spectrometer (NICOLET IR200, Thermo Scientific, Waltham, MA, USA). A 2.0 mg deposit of starch powder was blended with 200.0 mg of potassium bromide (KBr) powder and then compacted into disks. The infrared spectrum ranged from 4000.0–400.0 cm^−1^, and wavelengths were collected with scan times of 32.0 and a resolution of 4.0 cm^−1^.

#### 4.4.4. X-ray Diffraction (XRD) Analysis

XRD patterns of HBRSs were acquired by an X-ray spectrometry (XRD, SmartLab, Ragiku, Japan) with the diffraction angle 2*θ* of 3.0°–40.0° Bragg’s angle at a scanning speed of 8.0° (2*θ*)/min, and a step size of 0.02° (2*θ*). The relative crystallinities were assessed by MDI Jade 6 according to the previous method [53].

#### 4.4.5. Solubility, Swelling Power, and Water-Binding Capacity

The solubilities, swelling power, and water-binding capacities of HBRSs were assessed according to the previous report [20].

### 4.5. Simulated Gastrointestinal Digestion of the Modified Starches In Vitro

The simulated digestion was performed according to our previous study [54]. Next, 60.0 mg of HBRSs was dispersed in 20.0 mL of artificial digestion medium (89.6 g/L KCl, 20.0 g/L KSCN, 88.8 g/L NaH_2_PO_4_, 57.0 g/L Na_2_SO_4_, 175.3 g/L NaCl, 84.7 g/L NaHCO_3_, 2.0 g/L urea, and 290.0 mg α-amylase), and pH was adjusted to 6.8 by 0.1 M of HCl. The addition of 40.0 mL of distilled water to the medium started the salivary digestion and it was maintained at 37.0 °C for 5.0 min with a stirring speed of 150.0 rpm. Afterward, 10.0 mL of the digestion medium was collected and incubated in boiling water for 5.0 min to terminate the enzymatic digestion, which was regarded as the salivary digest before its pH was modified to neutral. The remaining medium was adjusted with 6.0 M of HCl at the final pH of 2.0. Then, 15.0 mg of pepsin was added to start gastric digestion. After incubating at 37.0 °C for 120.0 min, 10.0 mL of medium was collected, and a similar procedure was performed to obtain the gastric digests. Finally, the pH of the remaining medium was adjusted to 6.5 by 0.5 M of NaHCO_3_ and 5.0 mL of a mixture of pancreatin (8.0 mg/mL), lipase (6.0 mg/mL), and bile salts (50.0 mg/mL) was added to start the intestinal digestion. After another 120.0 min of incubation at 37.0 °C, the digestion medium was collected and deactivated to obtain the intestinal digests. Three replicated digestions were performed in each process independently. The free glucose released in digests was measured by a commercial kit and used to calculate the estimated hydrolysis ratio of starch.

### 4.6. Lipolysis Inhibitory Activity of the Modified Starches

#### 4.6.1. Digestion of Lipid Emulsion *In Vitro*

The lipid emulsion was prepared by dispersing sunflower oil in a simulated digestion medium at a volume ratio of 5:95 and stirring at 800.0 rpm for 12.0 h. After homogenization at 18,000.0 rpm for 3.0 min with 1.0 min intervals, a fine emulsion was obtained [55]. Then, 5 g of HBRSs was mixed with 25.0 mL of lipid emulsion and stirred at 150.0 rpm for 60.0 min. Based on the above procedure, the simulated digestion was operated. At each digestion phase, 0.5 mL of the digests were collected and used for the microstructure observation. Three independently replicated digestions were conducted for each sample. Cellulose was used as a control.

#### 4.6.2. Microstructure Analysis

A 0.5 mL of lipid emulsion collected at each digestion phase was stained by 10.0 μL of Nile red solution (5.0 mg/mL) for 5.0 min. Afterward, an aliquot of the stained emulsion was placed on the glass slide and covered with a coverslip. The microstructure of the emulsion was examined by a fluorescence microscope (Leica DM4 B, Leica microsystem GmbH, Wetzlar, Germany) at 20.0-fold magnification. The excitation and emission wavelengths were 543.0 nm and 605.0 nm, respectively. The droplet area of lipids was estimated by ImageJ software 1.52a (NIH, Bethesda, MD, USA).

#### 4.6.3. Release of Free Fatty Acids (FFAs)

After the intestinal digestion, the pH of the lipid emulsion was adjusted to 7.0 by a titrimetric method (0.05 M NaOH) using an auto-titrator (KLS-411, INESA scientific instruments, Shanghai, China). The consumed volume of NaOH solution was recorded to calculate the free fatty acids (FFAs) generated based on the molarity (mM) of oleic acid versus the amount of NaOH [56]. Assuming the generation of two FFAs per triacylglycerol molecule by lipase hydrolysis, the percentage of FFAs released was calculated as follows:$$\%\mathrm{FFAs}=\left(\frac{V_{NaOH}\times M_{NaOH}\times Mw_{Lipid}}{2\times W_{Lipid}}\right)\times 100$$
where *V_NaOH_* and *M_NaOH_* were defined as the volume (mL) and molarity of NaOH consumed. *Mw_Lipid_* and *W_Lipid_* were defined as the average molecular weight (876.16 g/mol) and initial weight of the sunflower oil [57].

### 4.7. Statistical Analysis

The differences between the individual groups were analyzed by one-way ANOVA with Duncan’s multiple-range tests. The results were presented as means ± SD, and values were considered statistically significant at *p* < 0.05. All statistical analyses were assessed using SPSS 23.0 software (SPSS Inc., Chicago, IL, USA).

## 5. Conclusions

In the current study, the physicochemical characteristics, digestibility, and lipolysis inhibitory potential of HBRSs prepared by autoclaving, microwave-assisted autoclaving, and enzymatic debranching were investigated and compared for the first time. It was found that HBSA and HBSM were characterized by their significantly higher swelling power and water-binding capacities, whereas their amylose contents and solubilities were distinctly lower than those of HBSI and HBSP (*p* < 0.05). Different modifications had little impact on the RS amounts among HBRSs. The enzymatic modifications of HBS improved the degrees of double helices and the ordered crystalline region in its structure, which restricted the susceptibility to digestion enzymes, contributing to its reduced digestibility, particularly for that of HBSP. In contrast, the physical modifications of HBS induced the formation of the amorphous region, which was beneficial for its swelling and binding with lipids, presenting relatively higher inhibitory effects on lipolysis. Additionally, structure-based morphology and particle size were also involved in the digestibility and lipolysis inhibitory activities of HBRSs. These results suggested that the physicochemical properties and functionalities of HBRSs primarily depended on their modification types.

## Figures and Tables

**Figure 1 molecules-28-01065-f001:**
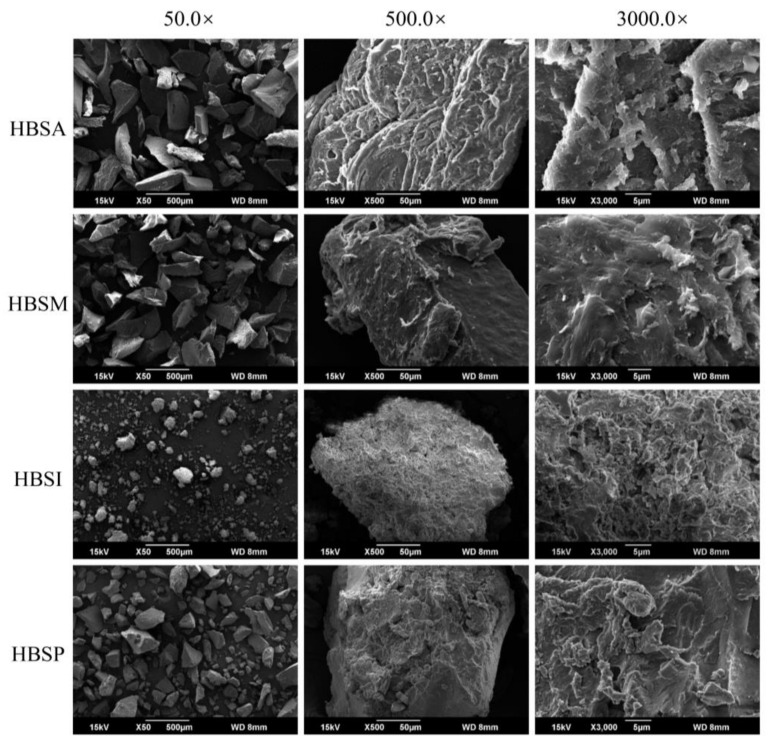
The morphological properties of HBSA, HBSM, HBSI, and HBSP.

**Figure 2 molecules-28-01065-f002:**
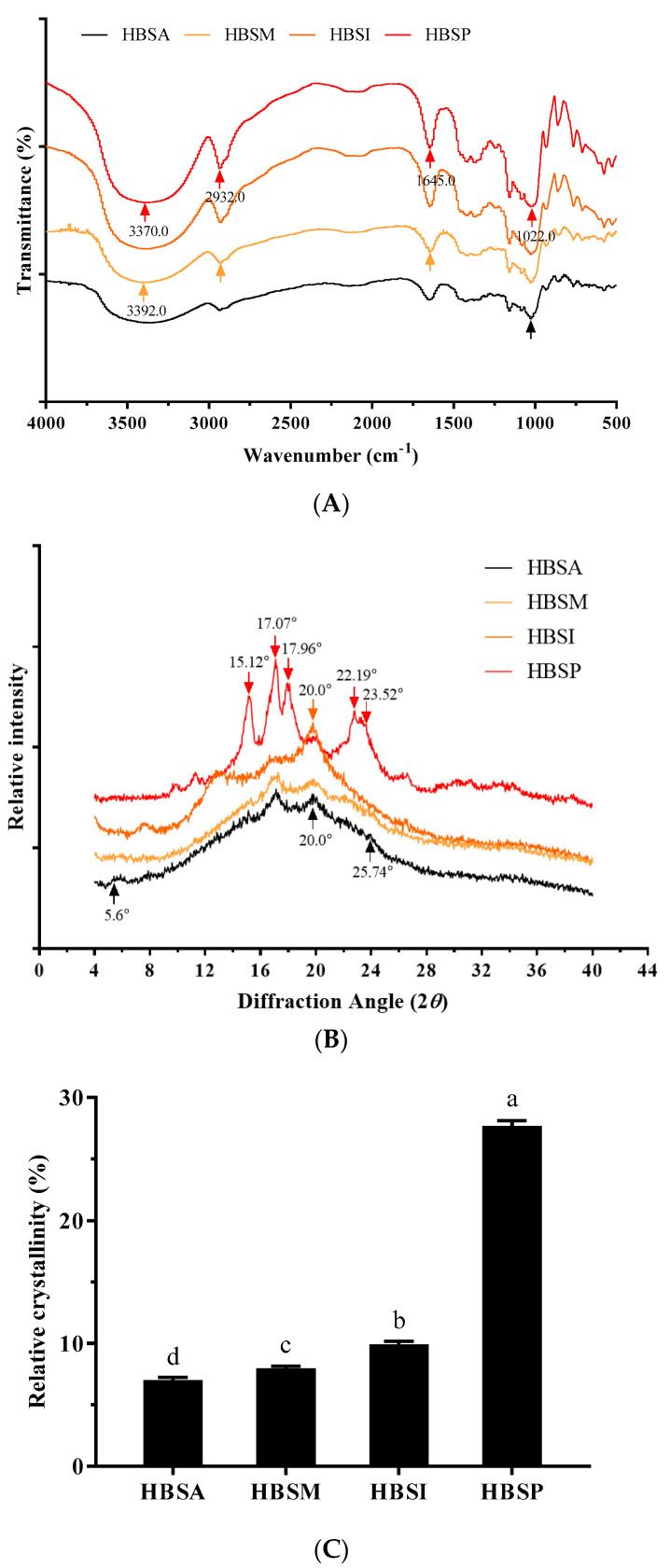
FT-IR spectra (**A**), XRD patterns (**B**), and relative crystallinities (**C**) of HBSA, HBSM, HBSI, and HBSP. Values with different lowercase letters were significantly different (*p* < 0.05).

**Figure 3 molecules-28-01065-f003:**
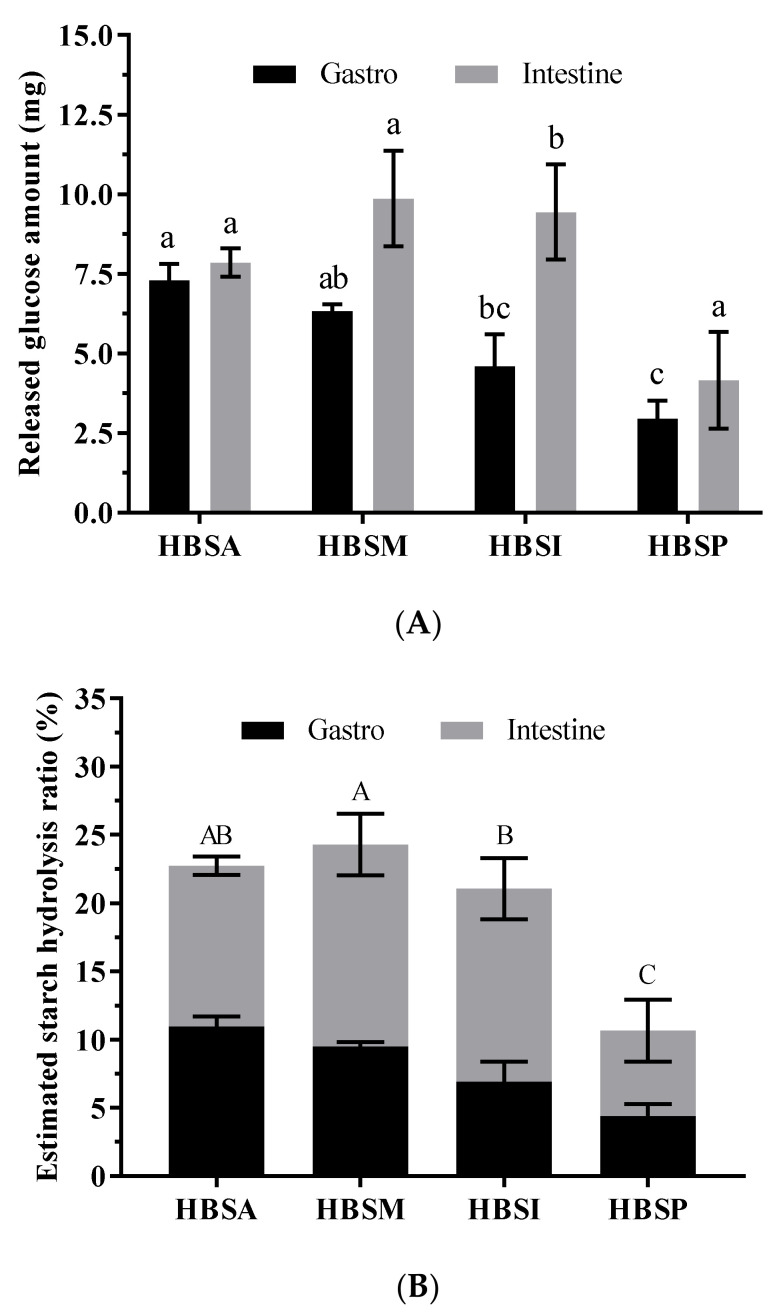
The released glucose contents (**A**) and estimated hydrolysis ratios (**B**) of HBSA, HBSM, HBSI, and HBSP during simulated gastrointestinal digestion. Values with different letters were significantly different (*p* < 0.05). Lowercase letters indicated differences among the released glucose amounts within the same digests. Capital letters indicated differences among the estimated starch hydrolysis ratios.

**Figure 4 molecules-28-01065-f004:**
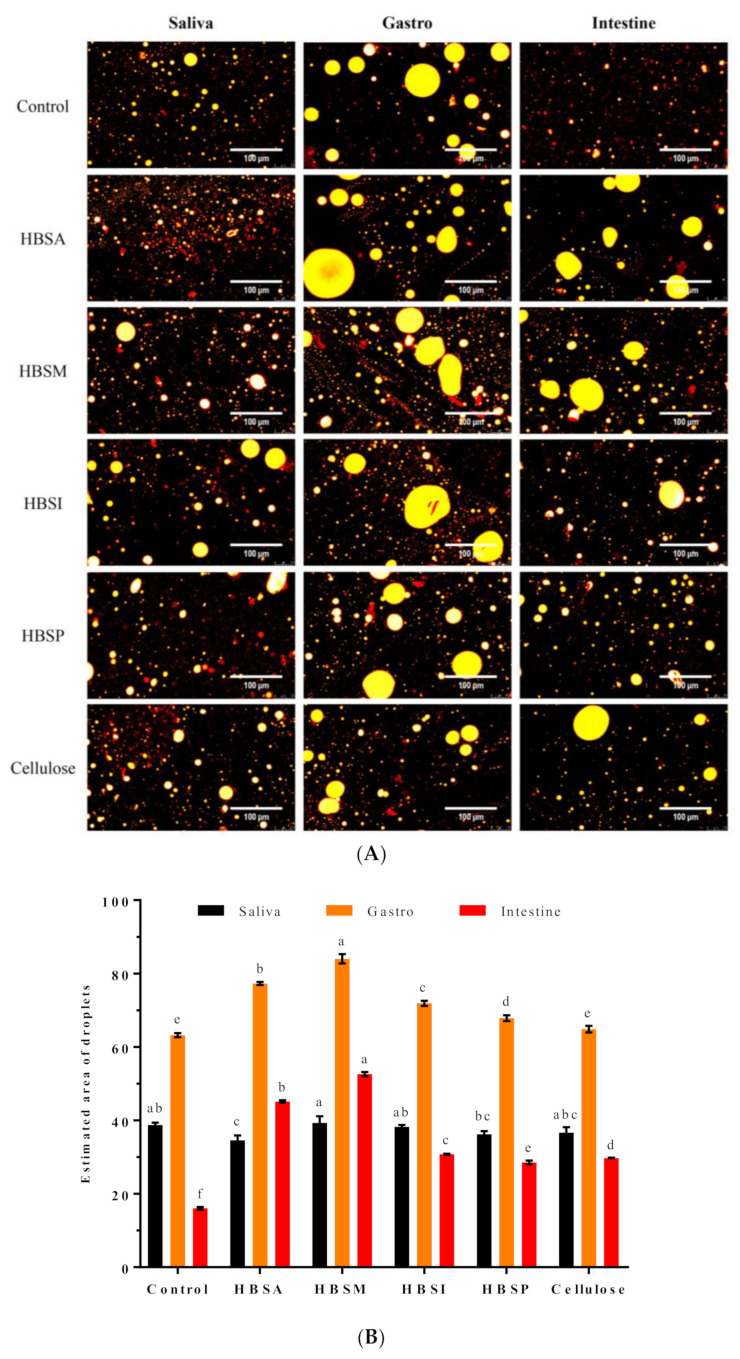
Effects of HBSA, HBSM, HBSI, and HBSP on microstructures (**A**), the estimated area of lipid droplets (**B**), and FFAs release rates (**C**) of lipid emulsion during lipolysis in vitro. Images were acquired at the magnification of 20.0× at room temperature. Values with different letters were significantly different (*p* < 0.05). Lowercase letters indicated differences among the EA of droplets within the same digestion phase. Capital letters indicated differences among FFAs release rates.

**Table 1 molecules-28-01065-t001:** Physicochemical characterizations of HBSA, HBSM, HBSI, and HBSP.

Samples	HBSA	HBSM	HBSI	HBSP
Amylose content (% ^1^)	24.53 ± 4.43 ^c^	29.25 ± 2.63 ^c^	51.29 ± 1.32 ^a^	36.02 ± 4.32 ^b^
Protein content (%)	nd ^2^	nd	nd	nd
RS content (%)	21.81 ± 0.22 ^a^	21.37 ± 0.48 ^a^	22.57 ± 1.96 ^a^	21.39 ± 0.89 ^a^
Solubility (%)	3.86 ± 0.07 ^b^	3.46 ± 0.25 ^b^	63.24 ± 0.01 ^a^	61.34 ± 2.87 ^a^
Swelling power (g/g)	16.96 ± 0.97 ^a^	14.58 ± 1.33 ^b^	9.58 ± 0.10 ^c^	11.51 ± 0.43 ^c^
Water-binding capacity (g/g)	2.51 ± 0.13 ^a^	2.52 ± 0.31 ^a^	1.37 ± 0.31 ^b^	1.48 ± 0.15 ^b^

^1^ % was expressed as the weight percent. ^2^ nd meant no data was detected. The data were presented as means ± SD (*n* = 3). Values with different lowercase letters in the same row were significantly different (*p* < 0.05).

## Data Availability

Data is contained within the article or Appendix A.

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
