# Peer review of "Physicochemical Characterizations, Digestibility, and Lipolysis Inhibitory Effects of Highland Barley Resistant Starches Prepared by Physical and Enzymatic Methods"

_molecules, 2023, doi:10.3390/molecules28031065_

Round 1
Reviewer 1 Report
Thank you for the opportunity to review this article.
The authors evaluated the Physicochemical characterizations, digestibility, and lipolysis inhibitory effects of highland barley resistant starches prepared by physical and enzymatic methods
The manuscript is generally well written in good order.
My overall recommendation is accept in the present form.
Author Response
Thanks very much for your recognition of our study, and the manuscript had been carefully re-checked to avoid minor flaws.
Reviewer 2 Report
The manuscript described physicochemical characterizations, digestibility, and lipolysis inhibitory effects in vitro of highland barley resistant starches prepared by physical and enzymatic methods. Some essential experiments were performed to support their statements and conclusions. Although a routine project, the work is well done and provide comprehensive evidence supporting the lipolysis inhibitory effects in vitro of four resistant starches. These results may be of particular interests of the broad readership and provided basis for preparation, characterization, and lipolysis inhibitory effects in vitro. Therefore, the submission can be accepted in Molecules after minor revision considering the following points:
Questions
1. Four kinds of resistant starch have been prepared in this paper, which one is the best? Please discuss this question in this paper.
2. The Authors should clearly indicate what is the scientific novelty of their research. Please indicate clearly what is new with your manuscript for the Molecules especially in comparison to earlier of publication(s).
3. The manuscript may be checked for grammatical and technical errors. Please modify the language to make it clear. For example, line 165, line 403, line 452, etc.
4. Please correct Figure 2A and 2B. FT-IR spectra (A), XRD patterns (B1).
5. The manuscript may be checked for some figure formats, including Figure 2~4,
6. Lines 226-227, I do not understand the values with different letters differed significantly. Please check whole the manuscript.
Author Response
- The manuscript described physicochemical characterizations, digestibility, and lipolysis inhibitory effects in vitro of highland barley resistant starches prepared by physical and enzymatic methods. Some essential experiments were performed to support their statements and conclusions. Although a routine project, the work is well done and provide comprehensive evidence supporting the lipolysis inhibitory effects in vitro of four resistant starches. These results may be of particular interests of the broad readership and provided basis for preparation, characterization, and lipolysis inhibitory effects in vitro.
Responses: Thanks very much for your recognition of our study, and the manuscript had been carefully revised as per your advice.
- Four kinds of resistant starch have been prepared in this paper, which one is the best? Please discuss this question in this paper.
Changes: Thanks very much for the suggestions. In the current study, the physical and enzymatic modifications were performed to improve the stability of native starch from highland barley (HBS) against digestion and endow its potential functionality. According to the results, the enzymatic debranched HBS revealed higher resistance to digestion than that modified by physical methods, particularly for pullulanase debranched HBS, which showed the lowest estimated hydrolysis ratio among highland barley resistant starches (HBRSs) (p < 0.05). In contrast, the lipolysis inhibitory activities of the physically modified HBS were significantly higher than those of the enzymatically debranched HBS (p < 0.05), as evidenced by their lower free fatty acids (FFAs) release ratios. Within the same modification type, microwave-assisted autoclaving modified HBS showed evidently higher inhibition on FFAs release than autoclaving modified HBS (p < 0.05), whereas the inhibitory activities of the enzymatically debranched HBS on FFAs releases differed slightly. Therefore, from the perspective of functionality, it could be concluded that the modifications of HBS by pullulanase debranching and microwave-assisted autoclaving were respectively favorable for reducing its digestibility and improving its inhibitory activity on lipolysis and the selection of modification method for HBS primarily depended on the desired targeted functionality. A detailed discussion had been added to the manuscript. Please check line 338-343 in the revised manuscript.
- The Authors should clearly indicate what is the scientific novelty of their research. Please indicate clearly what is new with your manuscript for the Molecules especially in comparison to earlier of publication(s).
Changes: Thanks very much for the suggestions. Growing evidence had demonstrated that highland barley (HB) was rich in functional ingredients, which were accountable for its superior health-promoting benefits over regular cereals. As the primary component of HB, starch was generally consumed as an energy supplier and was far from being investigated and developed compared with other components (e.g., polyphenols and beta-glucan) due to its high digestibility (rapidly digestive starch around 96.19 %) and low functionality. Promoting the transformation of native starch into resistant starch was recognized as a promising strategy for improving the stability and endowing novel functionality of native starch. In the current study, four modification techniques (physical and enzymatic methods) were carried out to prepare HBRSs, and the differences in their physicochemical characteristics and functionalities were assessed for the first time. The suppression of lipid digestion and absorption was significant for preventing hyperlipidemia and hypercholesterolemia. Previous studies had demonstrated the potential of resistant starch in managing lipid metabolism in vivo. The current study established a simulated digestion model for lipid emulsion digestion in vitro, providing straightforward evaluations of the inhibitory activities of HBRSs on lipolysis. Additionally, the association between the structural characteristics of HBRSs and their functionalities was further discussed and illustrated. The results of this study could facilitate the development of HBRSs as alternative functional foods and provide a basis for other starch modifications. The novelty of this manuscript had been introduced in the discussion. Please check line 245-246 in the revised manuscript.
- The manuscript may be checked for grammatical and technical errors. Please modify the language to make it clear. For example, line 165, line 403, line 452, etc.
Changes: Thanks very much for the suggestions. The grammatical and technical errors mentioned in line 165, line 403, and line 452 had been revised, the whole manuscript had been carefully re-checked, and the mistakes had been revised and marked with track changes.
- Please correct Figure 2A and 2B. FT-IR spectra (A), XRD patterns (B1).
Changes: Thanks very much for the suggestions. Figure 2A and 2B. FT-IR spectra (A), XRD patterns (B1) had been corrected and revised per your advice.
- The manuscript may be checked for some figure formats, including Figure 2~4.
Changes: Thanks very much for the suggestions. The formats of Figure 2~4 had been carefully checked and revised.
- Lines 226-227, I do not understand the values with different letters differed significantly. Please check whole the manuscript.
Changes: Thanks very much for the suggestions. The lowercase letters above the column in Figure 4 (A2, B) indicated the differences within the estimated areas of droplets and FFAs release rates among different intervention groups, with different letters representing significant differences based on the one-way ANOVA with Duncan’s multiple-range tests (p < 0.05). For clarification, the statistical description of Figure 4 had been modified as follows: “Values with different letters were significantly different (p < 0.05). Lowercase letters indicated differences among EA of droplets within the same digestion phase. Capital letters indicated differences among FFAs release rates.” Besides, the descriptions of statistical differences in other figures had also been carefully revised.
Reviewer 3 Report
In summary, the study found that autoclaving and enzymatic debranching methods were used to improve the stability and potential functionality of native starch from highland barley. It was found that the amylose content of the starch was significantly higher after these treatments and that this correlated with the formation of resistant starch. Microwave-assisted autoclaving was also tested but was found to not have a significant effect on the amylose content or resistant starch formation. The study also found that enzymatic debranching with isoamylase resulted in a higher amount of amylose than pullulanase debranching, indicating that the unit of -α-maltosyl maltotriose characterized the main structure of the starch. Additionally, the study found that the crystalline structure of the starch was altered by the different modification methods and that this affected the retrogradation process and the formation of resistant starch.
I recommend publication of the article in MDPI as I found research properly conducted and correctly presented conclusions.
Author Response

(The authors gave the same response as above.)
